# Application of Braided Piezoelectric Poly-l-Lactic Acid Cord Sensor to Sleep Bruxism Detection System with Less Physical or Mental Stress

**DOI:** 10.3390/mi15010086

**Published:** 2023-12-30

**Authors:** Yoshiro Tajitsu, Saki Shimda, Takuto Nonomura, Hiroki Yanagimoto, Shun Nakamura, Ryoma Ueshima, Miyu Kawanobe, Takuo Nakiri, Jun Takarada, Osamu Takeuchi, Rei Nisho, Koji Takeshita, Mitsuru Takahashi, Kazuki Sugiyama

**Affiliations:** 1Electrical Engineering Department, Graduate School of Science and Engineering, Kansai University, Osaka 564-8680, Japan; k99454567@kansa-u.ac.jp (H.Y.); k99484565@kansa-u.ac.jp (S.N.); k98554060@kansa-u.ac.jp (R.U.); k99554173@kansa-u.ac.jp (M.K.); k93440043@kansa-u.ac.jp (T.N.); k03456433@kansa-u.ac.jp (J.T.); 2Nishikawa Co., Ltd., Chuo, Tokyo 103-0006, Japan; sshimada@nishikawa1566.com (S.S.); ttnonomura@nishikawa1566.com (T.N.); 3Faculty of Foreign Language Studies, Kansai University, Osaka 564-8680, Japan; k95480112@kansa-u.ac.jp; 4Teijin Frontier Co., Ltd., Kita, Osaka 530-8605, Japan; nishior7@teijin-frontier.com (R.N.); takeshita-ko34@teijin-frontier.com (K.T.); 5Revoneo LLC, Fushimi, Kyoto 600-8086, Japan; contactmt@revoneo.com (M.T.); contactks@revoneo.com (K.S.)

**Keywords:** poly-l-lactic acid, piezoelectricity, braided cord, sensing, PLLA

## Abstract

For many years, we have been developing flexible sensors made of braided piezoelectric poly-l-lactic acid (PLLA) fibers that can be tied and untied for practical applications in society. To ensure good quality of sleep, the occurrence of bruxism has been attracting attention in recent years. Currently, there is a need for a system that can easily and accurately measure the frequency of bruxism at home. Therefore, taking advantage of the braided piezoelectric PLLA cord sensor’s unique characteristic of being sewable, we aimed to provide a system that can measure the frequency of bruxism using the braided piezoelectric PLLA cord sensor simply sewn onto a bed sheet on which the subject lies down. After many tests using trial and error, the sheet sensor was completed with zigzag stitching. Twenty subjects slept overnight in a hospital room on sheets integrated with a braided piezoelectric PLLA cord. Polysomnography (PSG) was simultaneously performed on these subjects. The results showed that their bruxism could be detected with an accuracy of more than 95% compared with PSG measurements, which can only be performed in a hospital by a physician and are more burdensome for the subjects, with the subjects simply lying on the bed sheet with a braided piezoelectric PLLA cord sensor sewn into it.

## 1. Introduction

Sleep affects humans in many ways; the lack of it causes fatigue, affects immunity, memory and learning, performance, and mental health, and most recently, causes dementia [1,2,3,4,5,6,7,8]. For mental health care, it is important to record daily sleep conditions and maintain and improve sleep duration and quality [9]. Polysomnography (PSG) is the gold standard for the objective assessment of sleep status [10,11]. However, PSG is a sophisticated diagnostic method, and the collection and accurate interpretation of results require specialized knowledge and skills in PSG measurement. The measurement system is also complex and expensive. Only a limited number of hospitals and laboratories offer PSG. Measurement during sleep requires the use of numerous sensors. Furthermore, the data obtained are visually analyzed by doctors [9,10,11]. Thus, the burden on the subject during measurement is great, and the greatest drawback is that measurement cannot be performed routinely at home. Therefore, with the growing interest in sleep in recent years, there is a need for a sleep tracker that can simply and routinely measure sleep status [9,10,11]. However, current sleep trackers are less accurate than PSG. With these as a background, our goal is to develop a device that can measure health status during sleep at home with an accuracy comparable to that of PSG without causing any burden or discomfort to the subject.

We previously conducted research using a braided piezoelectric poly-l-lactic acid (PLLA) cord [12,13], which has been attracting attention as a wearable sensor [14,15,16,17,18,19,20]. The braided piezoelectric PLLA cord sensor we have developed to date has many unique features. The following is a brief summary of the important features we have reported thus far [12,13]. First, plant-derived piezoelectric PLLA fibers are used as a motion-sensing material, and compared with other practical piezoelectric materials such as lead zirconate titanate (PZT), they do not contain heavy metals, such as lead or fluorine, and have less environmental impact [21,22,23,24,25,26,27,28]. Fibers are braided into a coaxial cable-like structure, making it resistant to electrical disturbances, as shown in Figure 1a. This structure of the piezoelectric PLLA braided cord was already reported [24,27]. The core of the piezoelectric PLLA braided cord is a conductive fiber bundle, and PLLA and PET fibers are wound around it. Furthermore, the conductive fibers cover them to realize a coaxial cable structure. The core was wrapped with PET fiber to form a braided piezoelectric PLLA cord. The cord is as mechanically strong as packing cords and is water-resistant. It can also be tied and untied due to its braided structure. On the other hand, PLLA fibers are monofunctional sensors that respond to bending motions and basically do not respond to stretching. However, if they are formed into, for example, a decorative knot to make an accessory-type sensor as shown in Figure 1b, they can respond to various motions [16,17,27,28]. This is a very significant feature of this braided piezoelectric PLLA cord sensor that PZT and other sensors do not have. The braided piezoelectric PLLA cord sensor can also be formed into various stitches with an embroidery needle. This is not only a design feature, but selectivity in sensing motion also can be achieved by embroidering a decorative knot or fabric [27]. For example, a choker with a lucky knot charm can detect only pulsation without being affected by the body’s motion even when the body is making a large motion. When chain stitches are embroidered on a denim fabric, only specific movements of each body part can be detected [12,13,19,20,27]. These results are supported by the findings of analysis with the finite element method (FEM), which identifies the bending displacement of the stitched braided piezoelectric PLLA cord sensor [12,13]. Therefore, taking advantage of the braided piezoelectric PLLA cord sensor’s unique characteristic of being sewable, we aimed to provide a system that can measure the frequency of bruxism by simply sewing the braided piezoelectric PLLA cord sensor onto a bed sheet.

In this study, we constructed and improved a system using a braided piezoelectric PLLA cord as a sensor. Then, the frequencies of bruxism in many subjects during one night of sleep were acquired using the sensor simultaneously with PSG measurement. The data thus obtained were compared to verify the accuracy of our system. As a result, we obtained a comparable accuracy to PSG. The results are reported below.

## 2. Difficulty in Measuring Bruxism

Bruxism is defined by the American Sleep Society as a “repetitive jaw muscle activity characterized by the clenching or grinding of teeth and/or the fixation or thrusting of the mandible” [29,30]. When teeth grinding occurs, the teeth are clenched hard and rub against each other repeatedly, which can aggravate gum sensitivity and periodontal disease [29,30,31,32,33,34,35,36,37,38]. It is also considered to cause temporomandibular joint disorder, facial pain, headaches, and stiff shoulders [31]. For those who sleep in the same room with others, bruxism can generate noise and also deteriorate the sleep quality of those in the same room. Stress and anxiety have been suggested as causes of bruxism [32], but a clear cause is not yet known. Treatment options are limited and include dental treatment and the use of a mouthpiece to prevent tooth wear [33,34]. There are two methods to diagnose bruxism: one is by interviewing subjects with abnormal dental conditions such as tooth wear caused by teeth grinding during sleep [9] and the other is using sensors such as those in PSG to detect bruxism [9,10,11,34,35,36,37,38]. In the former method, the only option is treatment because the diagnosis is made in the advanced state of symptoms. On the other hand, the method of directly detecting teeth grinding requires a device to be worn on the jaw, which is burdensome and cannot be used for daily measurement. These hurdles make it difficult to conduct research. When this diagnosis is conducted in the hospital, the subject wears the testing device and sleeps in a hospital bed overnight, and data are collected. Figure 2 below shows an illustration of this process. Sleeping in this state is stressful both physically and mentally due to the burden imposed by the testing equipment. In addition, since the examination equipment can only be used in a hospital, it is not possible to monitor the daily sleep status of a subject at home. Therefore, there is a need for a technology that can routinely monitor the condition of bruxism during sleep in a noncontact, nonburdensome manner. This would be useful in elucidating the causes, treatment, and prevention of bruxism. 

## 3. Braided Piezoelectric PLLA Cord Sensor

As a system for detecting the occurrence of bruxism that allows subjects to sleep soundly overnight without any psychological or physical stress, we considered integrating a braided piezoelectric PLLA cord sensor into a bed sheet. Changes in the subject’s sleeping posture cause major problems when detecting signals indicating the occurrence of bruxism for the following reasons. Originally, the braided piezoelectric PLLA cord sensor was based on the piezoelectricity of PLLA fibers. Piezoelectricity is a phenomenon that generates an electric charge in response to strain or stress applied to a material [39,40]. Therefore, if a bed sheet is subjected to a large amount of strain or stress due to body movement or tossing and turning during sleep, a large signal is generated on the basis of the piezoelectricity of the PLLA fibers. In other words, if a signal larger than the piezoelectric signal that would be generated by teeth grinding is generated by tossing and turning, it is superimposed on the signal generated by teeth grinding. The separation of these signals is expected to be difficult. In addition, there are various postures such as lying on one’s back or on one’s side. Piezoelectric sensors such as those constructed using PZT [39,40] are now in practical use. However, since the size of a PZT sensor is usually 3–5 cm, considering that the subject changes their lying position while sleeping, many PZT sensors must be spread over the entire bed and wired to each other. This is not a practical way when considering the time and effort required to do this. In contrast, a single braided piezoelectric PLLA cord can be easily sewn into a bed sheet over a large area that is responsive to changes in the posture of a subject lying on the bed sheet. The major problem here is the detection of the vibration generated by bruxism. The site of bruxism generation is considered to be around the jaw and mouth. However, the braided piezoelectric PLLA cord is sewn into the bed sheet. The braided piezoelectric PLLA cord is not directly in contact with the site of bruxism generation, but is rather in contact with the subject’s back and other parts below the neck. Common sense suggests that it would be difficult to detect the occurrence of bruxism with the braided PLLA piezoelectric cord sewn into the bed sheet under this condition. In previous studies, when such sensing was not possible, FEM was conducted to search for conditions under which sensing was possible [41,42], and a prototype sensor was successfully fabricated on the basis of FEM results. In this study, we followed the same approach and first conducted FEM to search for conditions under which the braided piezoelectric PLLA cord can be sewn into bed sheets to sense the vibration generated by bruxism.

### 3.1. FEM

The posture of the subject on the bed during sleep should be in a way such that the subject does not move away from the sensing area with the braided piezoelectric PLLA cord. Furthermore, considering that the way of contact with the braided piezoelectric PLLA cord changes depending on the subject’s posture, it is necessary to consider the method of sewing the braided piezoelectric PLLA cord. There are two main patterns of embroidering the braided piezoelectric PLLA cord on sheets. One is straight stitching, in which the braided piezoelectric PLLA cord is stitched perpendicularly to the fabric as if it were sewn with a regular sewing machine, and the other is zigzag stitching, in which the braided piezoelectric PLLA cord is placed on the fabric surface and fastened with a different thread. Since the site at which the subject comes in contact with the braided piezoelectric PLLA cord varies depending on the subject’s posture and sleeping position, it is important that the signal does not change at that time, which translates into system accuracy and simplicity. Therefore, we first investigated via FEM whether there is a difference in response between zigzag and straight stitching. Figure 3 shows the piezoelectric response of a model with the braided piezoelectric PLLA cord zigzag-stitched in a circle and applied with a stress of 10 N perpendicularly to the entire fabric. The color of the piezoelectric response indicates the magnitude of the response. The model with zigzag stitching shows almost the same piezoelectric response throughout the circumference. In other words, the piezoelectric response is the same regardless of the point of stress application on the circle. In contrast, as shown in Figure 4, the model with straight stitching shows a large piezoelectric response at the point where it touches the fabric and at the point of stress application on the fabric where the curvature of the folded braided piezoelectric PLLA cord changes. During sleep, the posture and position of the subject’s body vary from subject to subject, and even for the same subject, it varies from time to time. In other words, it is impossible to predict how the braided piezoelectric PLLA cord will come in contact with the subject’s body in this study. That is, it is strongly suggested that zigzag stitches, which generate the same piezoelectric response no matter where the braided piezoelectric PLLA cord comes in contact with the subject’s body, are suitable for the purpose of this study.

If we adopt zigzag stitching, the braided piezoelectric PLLA cord must be designed to have an inflection point that covers the entire bed sheet. To determine the effect of this design, the piezoelectric response was calculated for the braided piezoelectric PLLA cord having a curvature as shown in Figure 5. In particular, we paid attention to whether the piezoelectric response at the inflection point is much larger than that at other locations, as observed in straight stitching. As shown in the figure, a very detailed analysis of the calculation results shows that the piezoelectric response at the inflection point is indeed larger than those at other locations, but the rate of increase is less than 10%. From the calculation results, the final configuration of a single braided piezoelectric PLLA cord to be sewn onto a bed sheet was designed as shown in Figure 6 and Figure 7. Figure 6 shows that the cord covers a relatively large area of curvature where a constant piezoelectric response can be expected. On the other hand, the stitch pattern in Figure 7 has a longer-period curve than that in Figure 6. For Figure 6, a constant piezoelectric response is obtained. For Figure 7, the magnitude of the piezoelectric response is not affected by the addition of shorter-period curves. From these calculations, we decided to sew a single braided piezoelectric PLLA cord in a zigzag pattern to achieve a bed sheet with a configuration that provides a long-period curve.

### 3.2. Bed-Sheet Sensor Blueprint

The shape of the braided piezoelectric PLLA cord sewn in a zigzag pattern was determined from the results of FEM calculations. The specific size of the actual sheet sensor was determined from this shape, with particular consideration given to ensuring that the braided piezoelectric PLLA cord would always be in contact with the body in any position of the subject while lying in bed, even for a petite woman. To determine the size of zigzag stitching, human body dimensions were considered. Table 1 shows such data published by the Ministry of Economy, Trade and Industry, Japan. From this table, it can be determined that for the braided piezoelectric PLLA cord to be always in contact with the convexity of the body, even when a petite woman lies on her back, side, or at an angle in bed, the embroidery spacing of the braided piezoelectric PLLA cord must be 20 cm or less. Figure 8 shows a bed-sheet blueprint with the braided piezoelectric PLLA cord stitched in a zigzag pattern determined via FEM (hereafter, bed-sheet-type sensor). 

## 4. Bed-Sheet Sensor

From the FEM results, a design that fits the Japanese body shape was created, and a single braided piezoelectric PLLA cord was stitched in a zigzag pattern using a computerized sewing machine to make the stitches firm, as shown in Figure 9 (bed-sheet-type sensor). The sewn bed-sheet-type sensor was placed on the bed and covered with a mattress pad. In this system, which is the same as the previously reported system for the signal detection circuit, the sensed signal is received by a preamplifier and then amplified 400 times by an amplifier [41,42]. Basic measurements were conducted to confirm the responsiveness based on the piezoelectricity of the bed-sheet-type sensor. The sheet with a braided piezoelectric PLLA cord as the sensor sewn onto it was clamped at positions 10 cm to the left and right from the center of the bed sheet, and a static tension of 1 N was applied so that the sheet would not sag. An AC tensile strain with a frequency of 1 Hz and a distortion of 0.1% was applied, and a 700-fold amplified response signal was received. An example of the response signal is shown in Figure 10. The response waveform shows good reproducibility and continuity. Next, sheets were placed on actual beds used in sleep experiments. The sheet cover used in a sleep experiment was placed over another sheet to determine the pressure response. Deformation was applied by pulsing a 0.5 mm diameter circular brass rod pushed 0.5 mm into the bed-sheet-type sensor at the center. An example of the response is shown in Figure 11. The bed-sheet-type sensor was found to respond well to sharp pulses. 

## 5. Detection of Bruxism during Sleep

Subjects who agreed to participate in our experiment were fitted with PSG equipment during an overnight sleep at a sleep clinic. The experiment was conducted as follows. In this measurement, the bed-sheet-type sensor was placed on the bed where the subject slept, and PSG measurements were conducted simultaneously throughout the night. We emphasize here that the results obtained from the bed-sheet-type sensor in this experiment can be accurately contrasted with PSG results obtained under the supervision of a physician.

### 5.1. Medical Diagnostic Measurements

A brief description of the medical equipment used in the sleep clinic is given in [9,10,11] that focuses on features related to this experiment. PSG measurements (Figure 2) include electroencephalography (EEG) at Fp2-A1, F4-A1, C4-A1, and O2-A1; jaw and leg electromyography (EMG); bilateral electrooculography (BEOG); nasal airflow measurement; chest and abdominal respiratory movement detection; fingertip oxygen saturation measurement; electrocardiography (ECG); and a positional detection sensor fixed on the skin at the center of the sternum. Electrodes for EEG were placed on the head surface according to the international 10–20 method [9,10,11,34,35,36,37,38]. In addition, the body position during sleep was confirmed with an infrared camera. Specifically, EEG, BEOG, EMG of the jaw, ECG, abdominal movement detection, nasal airflow measurement, and oxygen saturation and snoring sound measurements were conducted. Video images and activity levels were also recorded. EEG and EMG of the jaw were conducted to determine tooth grinding [34,35,36,37,38]. Here, rhythmic masticatory muscle activity (RMMA) was determined by the technician on the basis of sleep stages, arousal, the visual assessment of movements, and EMG of the masseter muscle according to the AASM criteria [9,37]. The reason why the diagnosis of teeth grinding is so precise in such a sleep clinic is that it is based on the International Classification of Sleep Disorders [29,34,35,36,37,38], and sleep bruxism is considered to be one of the most common sleep disorders [9,37,38]. The appropriate processing of the data from the all-night EMG measurements in the hospital room was conducted by a clinical laboratory technician, and sleep bruxism was determined by a sleep specialist on the basis of the following criteria [9].
1)Mean amplitude of the electromyogram:More than 10% of the maximum occlusal force (masseter muscle) at waking time.2)Muscle contraction pattern during sleep bruxism episodes:(a)Phasic episode: 3 or more bursts (duration of 0.25 s to 2.0 s for each burst).(b)Tonic episode: one burst lasting more than 2 s.(c)Mixed episodes: bursts of both phasic and tonic episodes are present.

### 5.2. Bed-Sheet-Type Sensor Measurements

A mattress pad was placed on a sleep clinic bed for measurement. The bed-sheet-type sensor was placed directly on the mattress pad and fixed with pins to prevent it from shifting. The pad and the bed-sheet-type sensor were covered with a quick-drying, water-absorbent box sheet; thus, the bed-sheet-type sensor was not visible to the subject. The subject can sleep in any position, and our bed-sheet-type sensor does not restrict the subject’s sleeping posture. The subject lies down naturally with a pillow in the desired position, covers themselves with a blanket, and goes to sleep. The response signal from the bed-sheet-type sensor was amplified 400 times through a preamplifier and an amplifier and then stored in a data logger (NR-600B, Keyence corporation, Osaka, Japan; settings: 1 kHz and 12 bits) placed under the bed, as shown in Figure 12. The purpose of the circuit configuration is briefly explained below. First, since the impedance of the braided piezoelectric PLLA cord is very large, a preamplifier is used for impedance matching with the circuit. Furthermore, since this signal contains noise related to the power supply, Twin-T CR is used to remove the noise and amplify the weak signal for detection. Since the current measurement target is a human and the frequency bands of respiration, pulse, and body motion are 0.1 Hz to 10 Hz, the band-pass filter is used to detect these signals with high accuracy.The data obtained during sleep were stored overnight along with the PSG data described in the previous section.

## 6. Results

A demonstration experiment to show the effectiveness of the bed-sheet-type sensor sewn with a braided piezoelectric PLLA cord was conducted at a medical institution using a sleep test to diagnose sleep apnea syndrome. Note that the experiment at such a medical institution was conducted only when the physician had confirmed that the bed sheet embroidered with the braided piezoelectric PLLA cord as the sensor did not interfere with the diagnosis of sleep apnea syndrome and the subject consented to participating in the demonstration experiment.

### 6.1. Medical Judgment

In the experiment, subjects slept overnight in a clinic; they were fitted with PSG and other medical devices necessary to diagnose apnea, and they slept lying down on a bed. The bed-sheet-type sensor was placed under the bed cover. Before the measurement began, the subjects were first asked to lie down on the bed, and the medical staff examined them for any possible sleep disturbances caused by the bed-sheet-type sensor. All potential subjects responded that they felt no discomfort at all. Fifteen subjects participated in the demonstration experiment. During the overnight examination, some subjects experienced bruxism and others did not. An example of an actual all-night measurement of the waveforms during sleep obtained from the medical devices is shown. Figure 13 shows the ECG waveform, EMG waveform, activity levels, and postural changes. Note that EMG can detect weak signals at rest, but when body movements detected by the sleeping posture and activity meter (such as turning over) occur, EMG also detects a large response signal. On the other hand, ECG measures heartbeats continuously, and although the effect of body movements on ECG should not be large in terms of the measurement principle, body movements actually generate a large signal because the electrodes attached to the body change their state of adhesion. The amount of activity here refers to the total amount of activity per minute as measured by the physician with a small accelerometer attached to the subject’s waist. Since the amount of activity is constantly changing, the amount of activity in the figure is averaged over a 10 min period in order to show an overall trend. In the all-night measurement, a wide variety of signals are generated, which should not be the case in principle. It is very labor-intensive for technicians to individually determine sleep stages from the data obtained in such a complex environment. Overall, it can be seen in Figure 13 that the activity meter signal increases when the sleeping posture changes. In other words, the activity level increases at the timing of sleep turning. Sleep levels also change at this timing. At this time, a large signal is also generated in the ECG, which in principle should not be affected by turning over. Furthermore, determining the occurrence of teeth grinding is even more difficult. As mentioned earlier, the onset of bruxism is determined from changes in electromyographic signals in accordance with the aforementioned diagnostic rules [9,34,35,36,37,38,39]. The difficulty is that when the sleep duration is 6 h (21,600 s), a characteristic waveform lasting only 10–20 s is found in the data, from which the occurrence of bruxism is identified. For a subject who grinds their teeth, more than 50 episodes of bruxism occur in a single night. This indicates that even if such a complex PSG device could be fitted at home (which is not possible), it would be impractical to continue to observe the frequency of bruxism over a long period, even with current diagnostic methods, where a specialist must make the diagnosis.

### 6.2. Demonstration of Bed-Sheet-Type Sensor

To begin the analysis, the PSG signals when the subject was at rest and turned over and at all points where teeth grinding occurred were compared with the signals from the bed-sheet-type sensor. Representative results are shown in Figure 14, Figure 15 and Figure 16. In these figures, the magnitudes of the time and response signal axes are the same; thus, the magnitude and period of signals can be intuitively understood. As shown in Figure 14, when the subject was at rest, the signals from the bed-sheet-type sensor synchronized with the ECG signals. When the subject turned over in the bed, both the ECG and EMG signals were large, as shown in Figure 15. The signals from the bed-sheet-type sensor were also large. In the case of teeth grinding, the ECG signal was almost unchanged from the resting state, as shown in Figure 16. In the case of EMG, a high-frequency signal can be seen, although it is difficult to see on this scale (an enlarged image will be shown later). The high-frequency signal from the bed-sheet-type sensor also appears to be observed for a short period. From the above, a fast Fourier transform (FFT) process was applied to the overnight sleep signals to characterize the signal data obtained from the bed-sheet-type sensor during overnight sleep. Figure 17 shows a representative example of the signals obtained with the bed-sheet-type sensor for one night and the results of the FFT. In other words, these measurement data are unprocessed measurement signals that include all small signals, such as the vital signals of the subject and signals from body movements; the FFT results show that the detected signals are within a wide frequency range. In particular, the FFT results suggest that the absolute magnitudes of the signals are separated by frequency bands such as 0.1 Hz to 1 Hz, 1 Hz to 2 Hz, and 3 Hz to 7 Hz.

Using these results, primary filtering processing of the signals, low pass filter (LPF) processing (cut-off frequency: 0.5 Hz), band pass filter (BPF) processing (cutoff frequency: 0.8−1.5 Hz), and high pass filter (HPF) processing (cutoff frequency: 8 Hz) were performed to determine the characteristics of the signals from the bed-sheet-type sensor during the resting state, turning, and teeth grinding as determined by the physician. The results are shown below. First, representative results of the resting state are shown in Figure 18. LPF processing shows that the signal precisely synchronized with the respiratory signal, as shown at the bottom of Figure 18. The HPF-processed signal shows sharp pulses, indicating that it synchronized with the ECG signal, as shown at the top of Figure 18. Next, Figure 19 shows the data when the subject turned from lying on their belly to lying on their back. The top of Figure 19 shows the sleeping posture as determined by the physician. The activity level shows a peak when the sleeping posture changes. Here, the activity level is the sum of the amount of activity per minute. Thus, it can be seen that the activity level captures the change in sleeping posture well; the EMG signal is also larger, indicating that it is responding to the turning over. It also shows that the ECG which should not be affected by the measurement principle is also affected by a large amount. On the other hand, the bed-sheet type sensor responds from the beginning to the end of the turning over. In PSG of teeth grinding, ECG shows that the heart rate is unchanged as usual, as shown in Figure 20. Only EMG shows a characteristic signal. We sought to determine whether the bed-sheet-type sensor detects a characteristic signal. A typical example is shown below. 

Figure 21 shows EMG and HPF-filtered bed-sheet-type sensor signals at time when the doctor identified the occurrence of bruxism. It can be seen that the HPF-filtered bed-sheet-type sensor signal was different from the pulsation-based signal observed in the resting state shown in Figure 18a.

### 6.3. Algorithm for Bruxism Detection

As described above, it was found that the bed-sheet-type sensor can detect characteristic signals during teeth grinding; therefore, we constructed an algorithm for detecting the occurrence of teeth grinding using the bed-sheet-type sensor. To understand the characteristics of the waveforms obtained from the bed-sheet-type sensor, FFT processing was performed. The results are shown in Figure 22, which also shows the results in the resting state and when the subject turned over in bed. As can be seen, a large signal is generated below 1 Hz during turning over and other body movements. In contrast, when only grinding is occurring, the signal at frequencies below 1 Hz is small, as in the resting state. However, the signal at frequencies higher than 3 Hz is large only when teeth grinding occurs. This trend is compared with that of the results of EMG, which is used by physicians to determine teeth grinding. Figure 23 shows that the EMG and the bed-sheet-type sensor results show good agreement. Based on these results, the following processing flow was established to detect the occurrence of teeth grinding using only signals from the bed-sheet type sensor: (1) Real-time sensing is carried out by a bed-sheet-type sensor. (2) First-order HPF processing (cutoff frequency: 3 Hz) is applied to the sensor signals. (3) If the amplitude of the signal from the bed-sheet sensor is 5 times the average value over the past 20 min and continues for 2 s, a start flag is set. If the amplitude is less than 5 times the average value, an end flag shall be applied. (4) The FFT of the signal in the interval between the flags in (3) is calculated. The sum of the calculated amplitudes between 3 Hz and 7 Hz is obtained. When this sum is 100 times or more than the normal value (the average of the data from the time of falling asleep to the present), the occurrence of bruxism is judged.

In the physician’s evaluation, 3 of the 15 subjects were found to have no incidence of bruxism, whereas the other 12 subjects were found to have incidence of bruxism. For these 12 subjects, the occurrence of teeth grinding was independently determined from the bed-sheet-type sensor data using the algorithm described above. The results are summarized in Table 2. The following is an explanation of the table. For example, for subject No. 1, the physician identified the number of times teeth grinding occurred from the EMG waveform during one night of sleep, which was 150 times. On the other hand, the number of times teeth grinding occurred was independently identified by the above-mentioned process using the data of the bed-sheet-type sensor, which was 158 times. The number of times that the doctor’s judgment was consistent with the bed-sheet-type sensor was 150 times and the number of times of misjudgment was 8. In other words, every one of the 150 instances of teeth grinding identified by the physicians was precisely confirmed by the data from the bed-sheet-type sensors. It should be emphasized that the occurrence of teeth grinding was not missed at all by our bed-sheet-type sensor. However, the bed-sheet-type sensor results included eight false positives. We have carefully examined all the cases of misjudgment, and in all of them, a small body movement occurred for a moment. Table 2 shows that the average success rate for each subject when using the bed-sheet-type sensor in this study was over 90%.

Teeth grinding is a phenomenon that occurs in many people without their being aware of it, and its presence is usually only recognized when serious dental damage or sleep disturbances occur. Currently, the standard treatment for sleep disorders is to wear a mouthpiece, which many patients find physically and mentally stressful, thereby affecting their sleep quality. The widespread application of bed-sheet-type sensor systems for routine data collection has the potential to contribute significantly to advancements in research in this field. That is, the data routinely accumulated by bed-sheet-type sensor systems could facilitate the development of innovative therapies that are currently considered unfeasible. For example, it could lead to the development of specialized pillows and household products designed to facilitate sleep positions that prevent teeth grinding. The experimental results of the bed-sheet-type sensor system obtained in this study strongly suggest that the system may lead to the development of new treatments.

## 7. Conclusions

Bruxism is attracting attention as one of the factors that interfere with the maintenance and improvement of sleep quality, as it grinds and cracks teeth, aggravates gum sensitivity and periodontal disease, and causes temporomandibular joint disorder, facial pain, headaches, and stiff shoulders. In addition, for those sleeping together with others in the same room, the noise generated by bruxism worsens the quality of sleep of the other people. Thus, bruxism has a negative impact on health, and stress and anxiety have been suggested as some of the causes. However, a method of measuring it continuously in daily life has not been established. In this study, we developed a bed-sheet-type sensor consisting of a braided piezoelectric PLLA cord as a device that can measure bruxism without causing any burden or discomfort. This bed-sheet-type sensor offers a nonintrusive method of measuring bruxism, eliminating the need for direct body contact, unlike PSG. We have developed a device and an algorithm to identify and detect the unique waveform of bruxism. We used the device on subjects in a sleep clinic and determined the consistency of its results with judgments made by physicians on the basis of PSG results as a demonstration experiment. As a result, we obtained surprising results showing consistency with all the judgments made by the physicians. However, there were several cases in which the bed-sheet-type sensor system showed the occurrence of teeth grinding, but the physician judged that no teeth grinding occurred. In the future, we aim to fully implement the bed-sheet-type sensor system in society by enhancing collaboration with medical specialists, gathering more data from a diverse range of subjects, and refining the accuracy of its waveform analysis.

## Figures and Tables

**Figure 1 micromachines-15-00086-f001:**
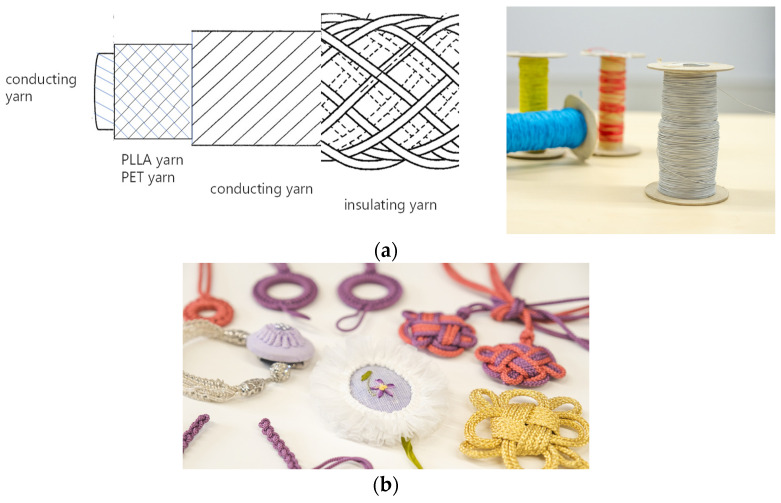
(**a**) Braided piezoelectric PLLA cord and (**b**) its decorative knots.

**Figure 2 micromachines-15-00086-f002:**
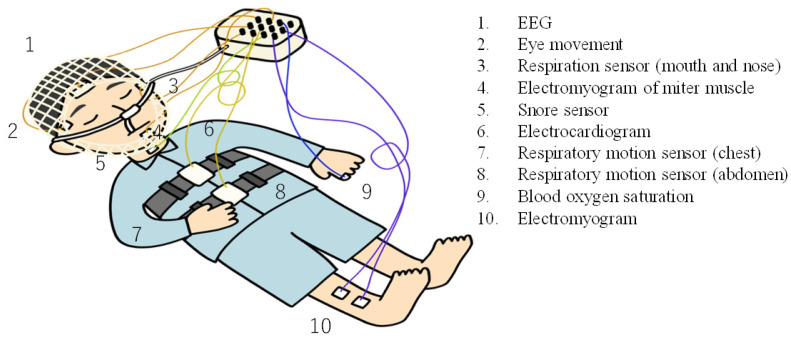
Illustration of PSG measurement during sleep.

**Figure 3 micromachines-15-00086-f003:**
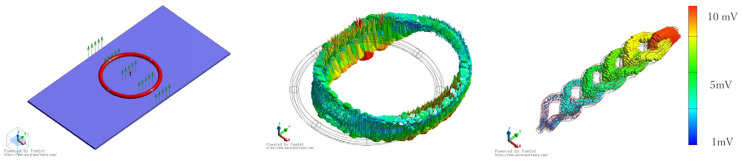
FEM calculation results of piezoelectric response of a model with braided piezoelectric PLLA cord zigzag-stitched in a circle.

**Figure 4 micromachines-15-00086-f004:**
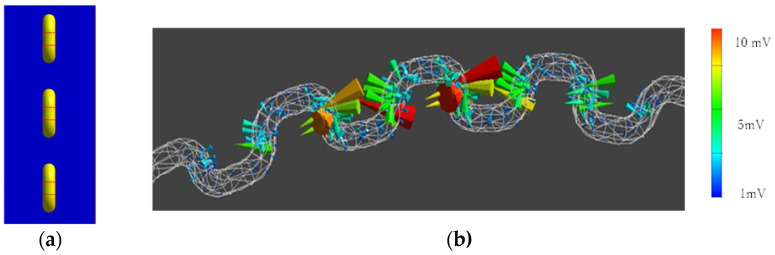
FEM calculation results of piezoelectric response of a model with braided piezoelectric PLLA cord stitched straight: (**a**) top view; (**b**) bird’s-eye view.

**Figure 5 micromachines-15-00086-f005:**
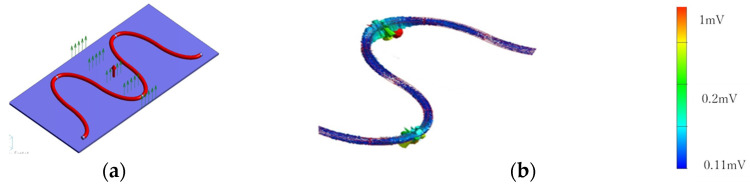
FEM calculation results of piezoelectric response of a model with braided piezoelectric PLLA cord stitched straight: (**a**) model with piezoelectric PPLA braided cord sewn onto fabric; (**b**) calculated values of response signal.

**Figure 6 micromachines-15-00086-f006:**
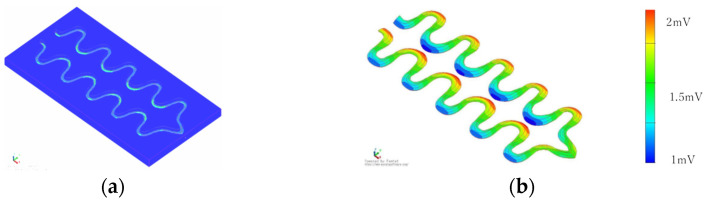
FEM calculation results of the configuration of the braided piezoelectric PLLA cord sewn onto the sheet (I): (**a**) model with piezoelectric PPLA braided cord sewn onto fabric; (**b**) calculated values of response signal.

**Figure 7 micromachines-15-00086-f007:**
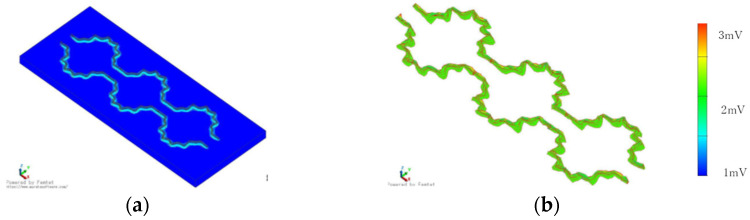
FEM calculation results of the configuration of the braided piezoelectric PLLA cord sewn onto the sheet (II): (**a**) model with piezoelectric PPLA braided cord sewn onto fabric; (**b**) calculated values of response signal.

**Figure 8 micromachines-15-00086-f008:**
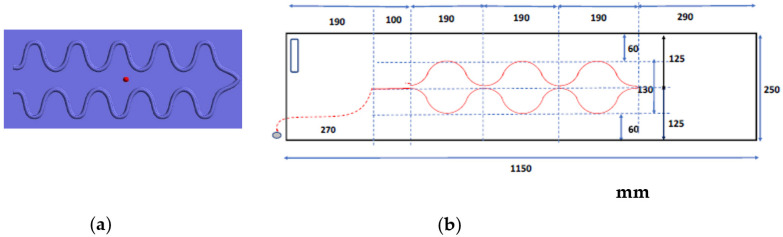
Illustration of the actual shape and dimensions of the sheet sensor prepared. (**a**) FEM model for bed-sheet type sensor. (**b**) Actual blueprint of bed-sheet-type sensor.

**Figure 9 micromachines-15-00086-f009:**
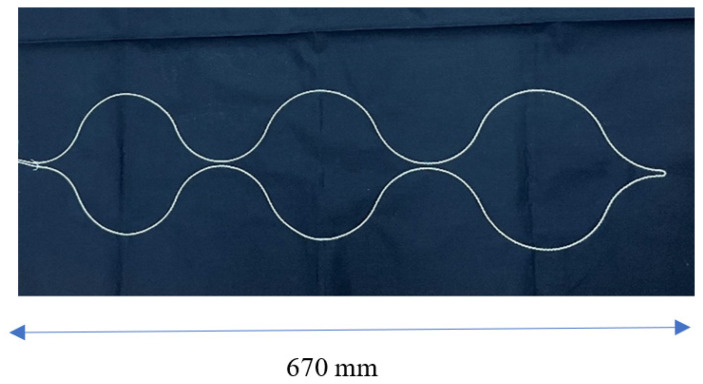
Photo of the completed bed-sheet-type sensor to be used in the experiment.

**Figure 10 micromachines-15-00086-f010:**
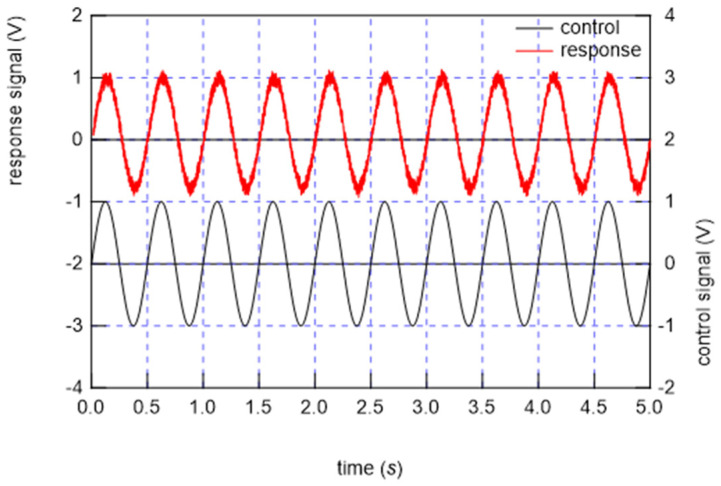
Example of response signal from the bed-sheet-type sensor when an AC tensile strain of 1 Hz frequency and a 0.1% strain rate were applied.

**Figure 11 micromachines-15-00086-f011:**
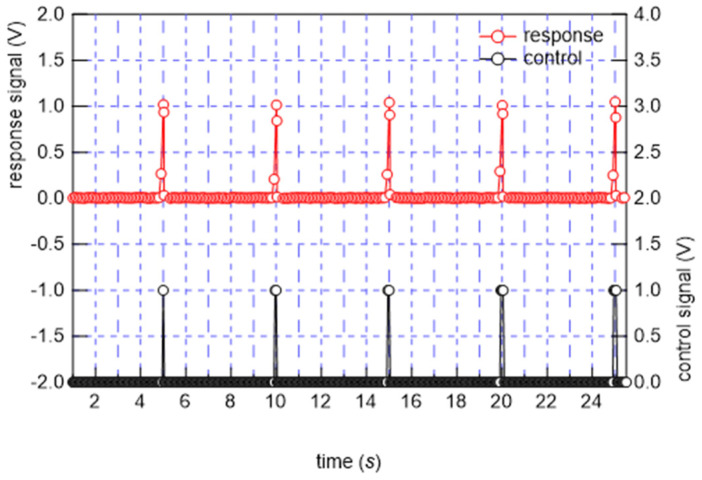
Response of bed-sheet-type sensor to the push-in displacement of the sensor.

**Figure 12 micromachines-15-00086-f012:**
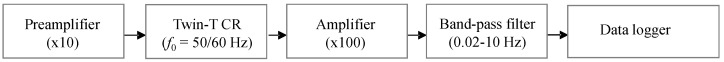
Circuit system for bed-sheet-type sensor.

**Figure 13 micromachines-15-00086-f013:**
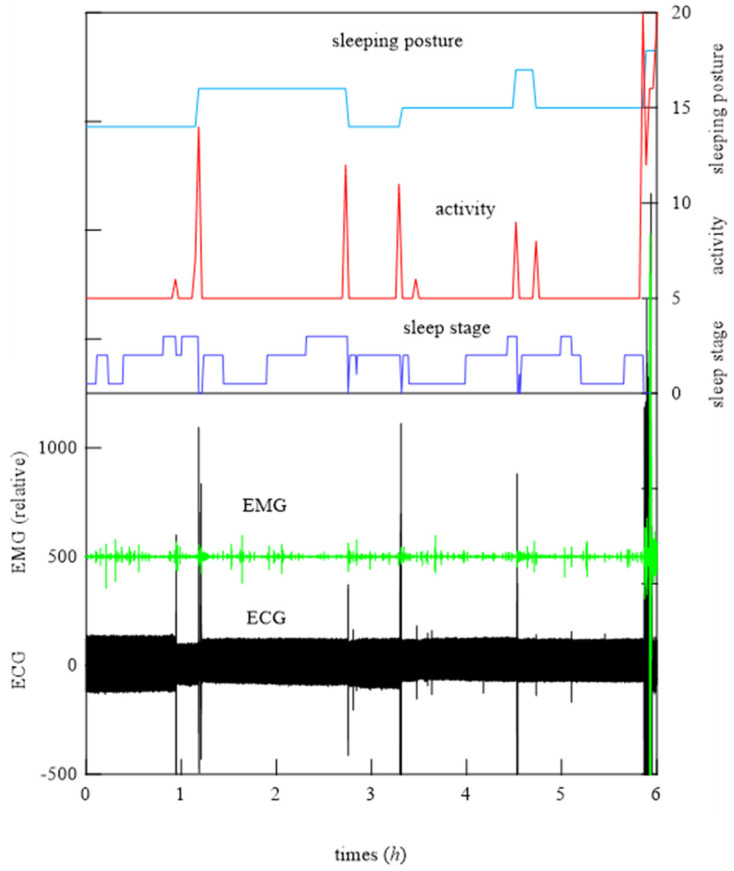
Example of ECG waveform, EMG waveform, activity levels, and postural changes during overnight sleep.

**Figure 14 micromachines-15-00086-f014:**
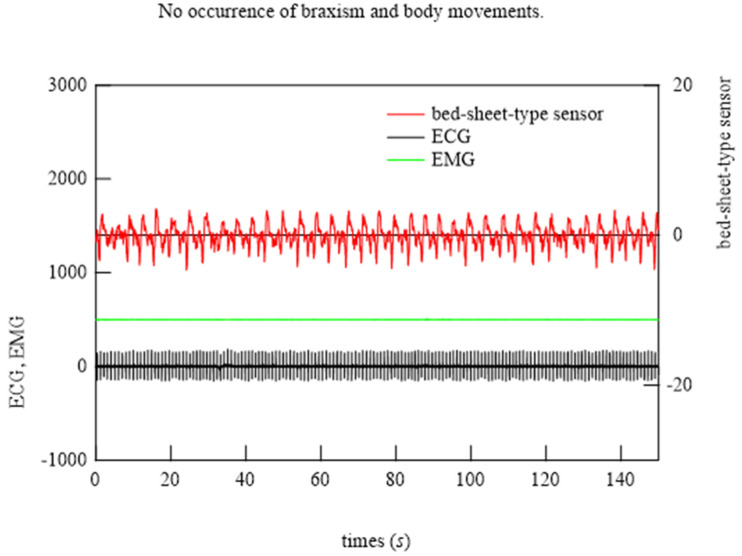
Typical examples of ECG, EMG, and bed-sheet-type sensor signals when the subject was at rest.

**Figure 15 micromachines-15-00086-f015:**
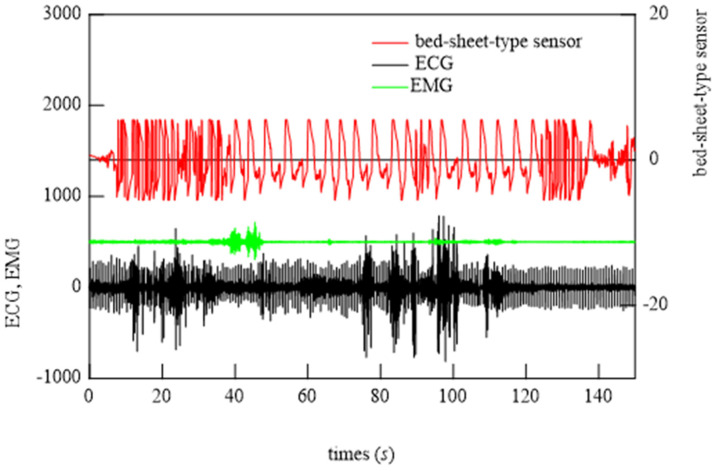
Typical examples of ECG, EMG, and bed-sheet-type sensor signals during turning over.

**Figure 16 micromachines-15-00086-f016:**
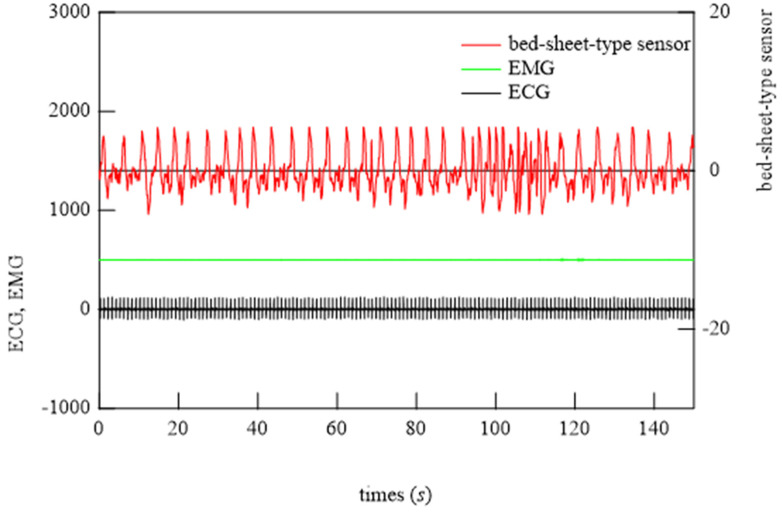
Typical examples of ECG, EMG, and bed-sheet-type sensor signals at the onset of bruxism.

**Figure 17 micromachines-15-00086-f017:**
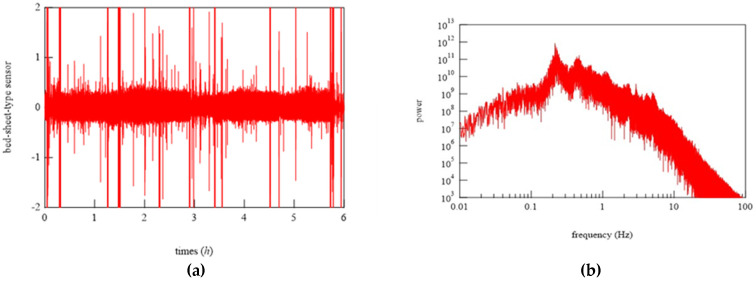
(**a**) Representative example of the signals obtained by the bed-sheet-type sensor for one night and (**b**) the results of the FFT.

**Figure 18 micromachines-15-00086-f018:**
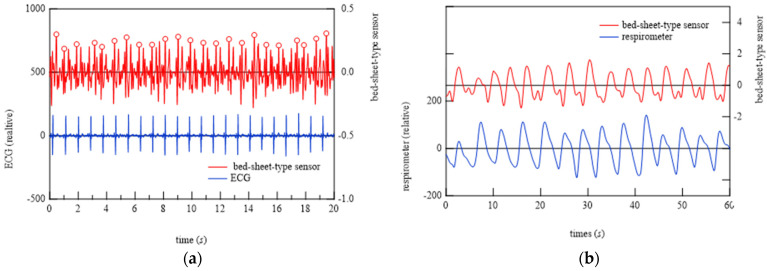
Response signals obtained from the bed-sheet-type sensor in this experiment obtained for (**a**) pulsation and (**b**) respiration.

**Figure 19 micromachines-15-00086-f019:**
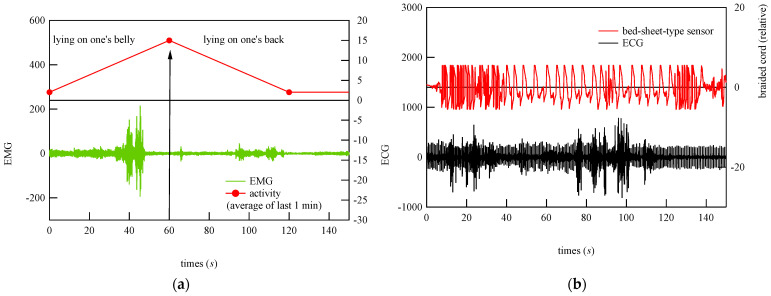
Typical example of simultaneous measurements with medical equipment (sleeping posture, activity, (**a**) EMG, and (**b**) ECG) in a hospital and with the bed-sheet-type sensor on a bed during a period that includes turning over in bed.

**Figure 20 micromachines-15-00086-f020:**
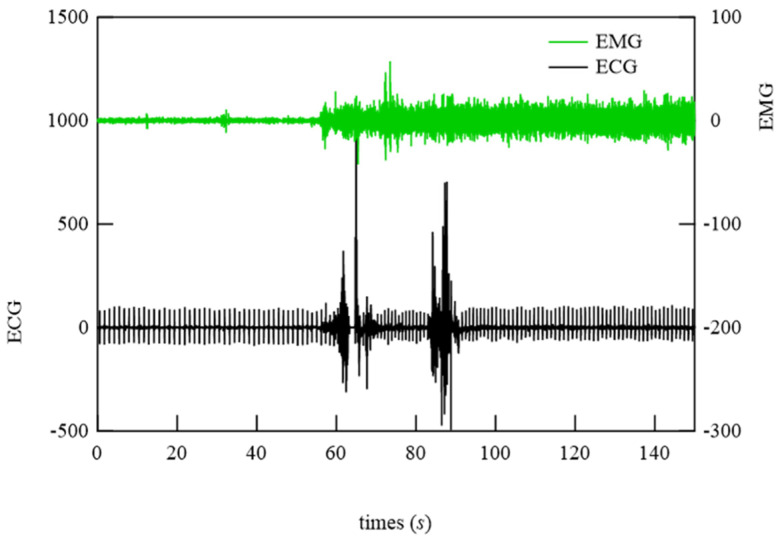
Two representative examples of EMG and ECG signals at the time of the physician-confirmed occurrence of bruxism.

**Figure 21 micromachines-15-00086-f021:**
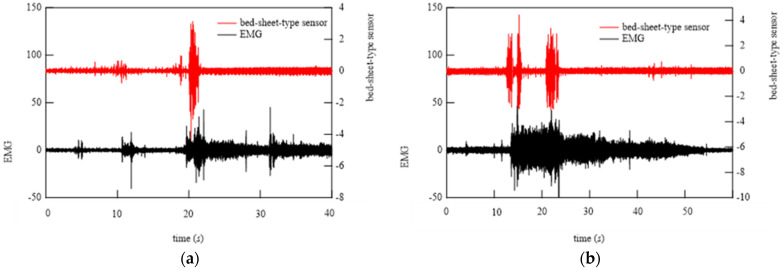
(**a**,**b**) Two representative examples of EMG signals and signals from the bed-sheet-type sensor at the time of the physician-confirmed occurrence of bruxism.

**Figure 22 micromachines-15-00086-f022:**
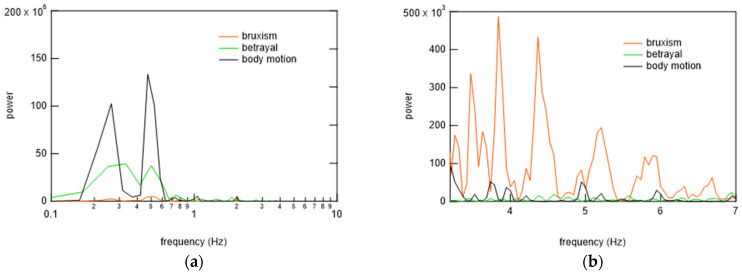
(**a**,**b**) Comparison of FFT signals of bed-sheet-type sensor during bruxism, turning over, and body movements.

**Figure 23 micromachines-15-00086-f023:**
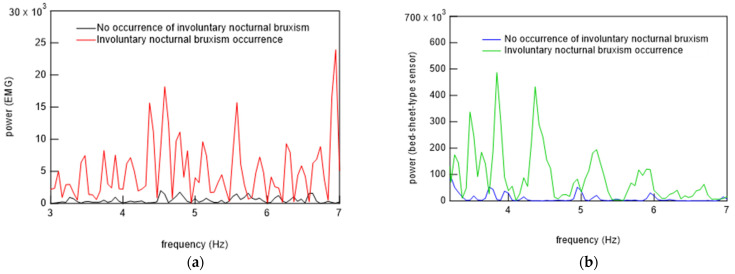
Comparison of FFT signals of (**a**) EMG and (**b**) bed-sheet-type sensor during bruxism and restful sleep.

**Table 1 micromachines-15-00086-t001:** Human body dimension measurement results (H19-10-1).

Distance between Right and Left Acromion (Not Necessarily a Straight Line)		Vertical Distance from the Acromion to the Lower End of the Elbow Bone Bent at a Right Angle		Horizontal Linear Distance between the Anterior and Posterior Surfaces of the Chest at the Nipple Point	
male	female	male	female	male	female
mm		mm		mm	
403	358	340	338	201	201
404	360	315	309	212	200
406	358	341	335	222	200
406	359	316	308	225	204
404	360	342	329	228	209
403	359	314	307	230	210
399	358	341	329	231	214
395	359	314	306	229	219
391	356	339	330	225	222
388	352	312	305	228	229
385	350	339	328	230	231
380	347	309	300	228	233
From the Ministry of Economy, Trade and Industry Japan					

**Table 2 micromachines-15-00086-t002:** Comparison of diagnostic and bed-sheet-type sensor judgment results of bruxism.

Subject	Gender	Age	Height (cm)	Weight (kg)	BMI	Medical Diagnosis	Bed-Sheet-Type Sensor			
							Determination by Bed-sheet-Type Sensor	Number of Times of Correct Judgment	Number of Times of Misjudgment	Success Rate (%)
1	male	34	163	96	36	150	158	150	8	94.9
2	female	81	159	58	23	30	35	30	5	85.7
3	male	39	173	67	22	51	53	51	2	96.2
4	male	35	155	60	25	34	38	34	4	89.5
5	male	15	164	49	18	46	54	46	8	85.2
6	male	47	181	68	21	88	94	88	6	93.6
7	male	46	170	90	31	91	97	91	6	93.8
8	male	58	170	64	22	93	99	93	6	93.9
9	female	38	170	82	28	41	47	41	6	87.2
10	male	61	167	69	25	74	82	74	8	90.2
11	male	64	175	98	32	47	49	47	2	95.9
12	female	58	163	66	25	47	49	47	2	95.9
average of success rate (%)										91.9

## Data Availability

Data are contained within the article.

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
