# Peer review of "Application of Braided Piezoelectric Poly-l-Lactic Acid Cord Sensor to Sleep Bruxism Detection System with Less Physical or Mental Stress"

_micromachines, 2023, doi:10.3390/mi15010086_

Round 1

Reviewer 1 Report

Comments and Suggestions for Authors

The authors investigated a bed-sheet-type sensor consisting of a braided piezoelectric PLLA cord as a device that can measure bruxism without causing any burden or discomfort, which offers a non-intrusive method of measuring bruxism, eliminating the need for direct body contact, un-like PSG. The authors have developed a device and an algorithm to identify and detect the unique waveform of bruxism. The reviewer thinks that researchers in this field would be very interested in the contents of this manuscript. Therefore, the reviewer would recommend the publication of this paper to Micromachine after the following minor revision. The following comments are not to criticize but to strengthen the manuscript.

- There is no figure in the text that provide specific descriptions of materials and devices. The reviewer suggest the author to add a figure explaining how the material was synthesized and the device was made.

- Is the PLLA cord shown in the manuscript an intrinsically stretchable material? If so, is there any reason to need a zigzag structure? If it is not an intrinsically stretchable structure, what advantage does this structure have over an intrinsically stretchable structure?

- The reviewer ask the authors to explain why a preamplifier, amplifier, and band-pass filter are needed in a measurement system in detail?

- Is the system presented by the author commercializable? I think it would be a good idea to write down the level at which commercialization is possible.

Comments on the Quality of English Language

Author Response

 Dear Reviewer1,

Thank you for your valuable feedback and suggestions regarding our manuscript. We have carefully considered your comments and have made the following revisions to our paper:

>There is no figure in the text that provide specific descriptions of materials and devices. The reviewer suggest the author to add a figure explaining how the material was synthesized and the device was made.

This is important information, and I appreciate your guidance. Following your suggestion, I have added Figure 1 and expanded the explanation in the relevant section (lines 63-68).

>Is the PLLA cord shown in the manuscript an intrinsically stretchable material? If so, is there any reason to need a zigzag structure? If it is not an intrinsically stretchable structure, what advantage does this structure have over an intrinsically stretchable structure?

As you correctly noted, this enhancement increases the sensor's elasticity. Thank you very much.

>The reviewer ask the authors to explain why a preamplifier, amplifier, and band-pass filter are needed in a measurement system in detail?

Your insight is greatly valued. We have incorporated additional information in lines 277-285 in response to your comment

->Is the system presented by the author commercializable? I think it would be a good idea to write down the level at which commercialization is possible.

Thank you for your valuable feedback. We are confident that the basic performance of our product will meet your expectations. However, we have not yet commenced practical application tests, including those for durability. In collaboration with medical institutions, we are committed to conducting further experiments to advance our product towards practical application.

We believe that these changes have significantly improved our manuscript, making the arguments clearer and the research more robust. We appreciate your help in enhancing the quality of our work.

Thank you once again for your insightful feedback and for the opportunity to improve our manuscript.

Sincerely,

Prof. Y. Tajitsu. Ph.D.
Kansai University

Reviewer 2 Report

Comments and Suggestions for Authors

In this paper, the authors present a braided piezoelectric PLLA cord sensor that enables the detection of teeth grinding symptoms at home. The proposed sensors in this paper is meaningful to human health. However, the presentation of the research should be modified. Following are some issues for the authors.

1.      The title of the manuscript is too long, and it should be simplified.

2.      There are too many figures in the manuscript. To enhance the readability, it is recommended to merge part of the figures, and put some figures into a supporting information.

3.      Figures and their caption should be modified. For example, in Figure 8, The unit is missed; In Figure 9, it is recommended to add a scale bar to inform the size of the sensor; In Figure 13, what is the meaning of the activity levels? And the time resolution of the activity levels appears to be very low.

4.      In Figure 19, what is the meanings of the blue region and the yellow curve in the top part, how they indicate the state of a person in different postures and under different activities. In addition, the bottom part is not mentioned in the manuscript.

5.      In lines 378 to 384 the authors describe the algorithm for determining whether or not a tooth grinding symptom has occurred. The reference to the average value in step (3) is unclear, does it mean the average value of the amplitude between the start and that moment or the average value obtained by other tests. Second, the article states that the start flag is set if the large amplitude signal lasts for two seconds. Please explain how this duration is follows.

6.      In the abstract, the authors state that teeth grinding can be detected by the braided piezoelectric PLLA cord sensor with an accuracy of 95% or more, but in the data given in Table II, it appears that only three volunteers achieved this goal, with an overall average accuracy of only 91.9%. The authors are requested to double-check.

Author Response

Dear Reviewer2,

Thank you for your valuable feedback and suggestions regarding our manuscript. We have carefully considered your comments and have made the following revisions to our paper:

  1. The title of the manuscript is too long, and it should be simplified.

Following your suggestion, we have shortened the overly lengthy title. Thank you very much.

  1. There are too many figures in the manuscript. To enhance the readability, it is recommended to merge part of the figures, and put some figures into a supporting information.

Per your suggestion, we have revised Figures 8, 9, and 19, which was notably challenging to understand. Thank you very much.

  1. Figures and their caption should be modified. For example, in Figure 8, The unit is missed; In Figure 9, it is recommended to add a scale bar to inform the size of the sensor; In Figure 13, what is the meaning of the activity levels? And the time resolution of the activity levels appears to be very low

Thank you for your suggestion. We have added units to Figure 8, included a scale in Figure 9, and provided additional explanation for Figure 13 (lines 310-321).

  1. In Figure 19, what is the meanings of the blue region and the yellow curve in the top part, how they indicate the state of a person in different postures and under different activities. In addition, the bottom part is not mentioned in the manuscript.

My apologies for the previously unclear text. I have revised the text in lines 363 to 370 and included the figure, specifically.

  1. In lines 378 to 384 the authors describe the algorithm for determining whether or not a tooth grinding symptom has occurred. The reference to the average value in step (3) is unclear, does it mean the average value of the amplitude between the start and that moment or the average value obtained by other tests. Second, the article states that the start flag is set if the large amplitude signal lasts for two seconds. Please explain how this duration is follows.

We apologize for the confusion. A significant revision has been made from lines 392 to 402.

  1. In the abstract, the authors state that teeth grinding can be detected by the braided piezoelectric PLLA cord sensor with an accuracy of 95% or more, but in the data given in Table II, it appears that only three volunteers achieved this goal, with an overall average accuracy of only 91.9%. The authors are requested to double-check.

My sincerest apologies for this oversight. We have corrected the error and have also thoroughly reviewed the remaining sections to ensure accuracy. We deeply regret any inconvenience caused.

We believe that these changes have significantly improved our manuscript, making the arguments clearer and the research more robust. We appreciate your help in enhancing the quality of our work.

Thank you once again for your insightful feedback and for the opportunity to improve our manuscript.

Sincerely,

Prof. Y. Tajitsu. Ph.D.

Kansai University